# A Multi-User Collaborative Access Control Scheme Based on New Hash Chain

Zetian Wang [1,†], Yunfa Li [2,*] , Guanxu Liu [2,†] and Di Zhang [2,†]

1 Key Laboratory of Complex Systems Modeling and Simulation, HDU-ITMO Joint Institute, Hangzhou Dianzi University, Hangzhou 310018, China
2 Key Laboratory of Complex Systems Modeling and Simulation, School of Computer Science and Technology, Hangzhou Dianzi University, Hangzhou 310018, China
* Correspondence: yunfali@hdu.edu.cn
† These authors contributed equally to this work.

**Abstract:** As the threats to the Internet of Things (IoT) continue to increase, access control is widely used in various IoT information systems. However, due to the shortcomings of IoT devices such as low computing power, it is impossible to use high-performance methods to control user access. Although the emergence of the blockchain provides another way of thinking for access control, the implementation based on the blockchain requires the device to complete the proof of work (PoW) and requires the device to have high computing power. At the same time, most access control schemes existing today are intended for users to use alone, which cannot be applied to the field of multi-user coordinated access. Therefore, this paper proposes a multi-user collaborative access control scheme based on a new hash chain, which uses the identity information of multiple users as the seed value to construct the hash chain, and uses the hash chain as the PoW of the blockchain. An efficiency analysis showed that this method requires only a small amount of hash value calculation and can be applied to IoT systems with low computing power. The security analysis shows that the scheme can resist a variety of attack methods and has high security.

**Keywords:** blockchain; IoT; hash chain; access control; proof of work

## 1. Introduction

Nowadays, the IoT is developing rapidly. More and more people have begun to use IoT services in their lives, which means more and more user data exist in IoT devices and networks. However, at the same time, attacks against IoT devices and networks are becoming more and more frequent, and users' privacy data are often leaded. Therefore, how to solve the security risks existing in IoT services and improve the security of user privacy data has become a hot research issue.

As the entrance of the IoT system, access control can effectively intercept illegal users and authorize legitimate users to access their own data, which is a means of making IoT services more secure. Traditional access technologies mainly include password-based authentication, biometric-based authentication, and certificate-based authentication. Among them, the biometric-based authentication develops particularly rapidly. Especially with the development of recognition technology, the accuracy of authentication has been continuously improved. For example, the new feature pyramid composite neural network structure [1] proposed by Xiao et al. can enhance the context, thereby improving the performance of small target detection, and can be well applied in this field.

However, most of these traditional access methods are based on a centralized architecture, which may cause problems such as a single point of failure. If it is attacked by a network, this will easily lead to the paralysis of the entire system. Additionally, with the development of IoT applications, some data are no longer only for individuals, but belong to multiple users, which requires the access system to realize multi-user coordinated access.

However, most of the traditional access solutions are based on single-user management, and cannot access multiple users at the same time.

The emergence of the blockchain provides another way of thinking for IoT identity access. Ma et al. [2] proposed a blockchain-based method, which forms a blockchain network through edge devices, which can provide secure access control in the IoT environment, and the method can also tolerate errors. In blockchain applications, users can use the data without knowing the content of the shared data. Therefore, the blockchain can be used to save the user's verification data, and it can complete the access control without knowing the specific content of the verification data. Therefore, more and more researchers have begun to study the feasibility of using blockchain for identity access.

However, there are certain limitations to using the blockchain to achieve access. Because the blockchain requires the equipment to complete the PoW, which often requires the equipment to have high computing power, it undoubtedly increases the cost of building an IoT system. At the same time, because IoT devices need to spend a lot of time on calculating and solving, this not only affects the efficiency of access, but may also continue to consume the life of the devices. In addition, the existing blockchain technology also focuses on single-user access, which cannot meet the needs of multi-user coordinated access.

The hash chain continuously hashes the message to generate hash values through specific rules, so the receiving end can verify the message by using the same rules to build a hash chain. At the same time, due to the irreversible nature of the hash chain, it can be well combined with the PoW in the blockchain. In 2020, Kim et al. [3] propose using the reverse hash chain as the PoW. This scheme allows the user holding the key to quickly complete the PoW, thereby completing the authentication. However, their method only supports a single user; because the hash chain is only a continuous iteration of the seed value, it is not suitable for the scene of multi-user coordinated access.

Therefore, in order to achieve multi-user collaborative access control and solve the limitations of blockchain access control, this paper proposes a PoW method based on a new hash chain. This method combines the identity information of multiple users in a special way and hashes iteratively, and finally generates a new type of hash chain for PoW. At the same time, this paper proposes a multi-user cooperative access control method based on the new hash chain. This method uses the new hash chain to complete the PoW, and finally realizes the multi-user cooperative access control.

## 2. Related Work

### 2.1. Access Control

A key agreement scheme is firstly used in user access control. However, many flaws are found in the key agreement scheme. Tai et al. [4] found the security flows in the scheme and proposed improvement plans. There are many people also using the blockchain to control access. Practical Byzantine Fault Tolerance (PBFT) is an algorithm that can tolerate the errors of a few nodes and obey the majority of nodes. Based on the PBFT, Gong et al. [5] proposed a gateway for recording access control in the blockchain. Using a public dataset, Gong evaluated the accuracy of the model and the accuracy of Fraud Detection. Huang et al. [6] proposed a useful system for controlling access; the method they proposed is mainly based on the blockchain, and the main purpose is to improve the efficiency and security of the IoT server architecture. Wang et al. [7] proposed an access control protocol, which is mainly used for IoT terminal devices. The protocol is used to protect privacy, and its structure is adaptable. Huang et al. [8] proposed a framework for access control, which can provide protection for embedded devices. Takieldeen et al. [9] proposed an access control pipeline, which combines three schemes, namely the ECC scheme, the OSS signature scheme and the chaotic mapping scheme. Luo et al. [10] proposed a framework to manage identities hierarchically, and they also proposed a combined authentication and session key protocol. Xiang et al. [11] proposed a PBBIMUA scheme, which is based on a blockchain for user authentication and identity management. Jia et al. [12] proposed a key agreement protocol, which is an identity-based anonymous authentication

protocol, mainly used in the MEC environment. Cui et al. [13] proposed a blockchain-based multi-WSN authentication scheme for IoT. Tsai et al. [14] proposed an efficient distributed authentication scheme, which uses cloud computing services, and can allow mobile users to use cloud computing services securely and conveniently. Wang et al. [15] proposed a rapid authentication scheme based on identity, which is mainly suitable for smart mobile devices in WBANs. Fan et al. [16] introduced the blockchain and the identity authentication method using the blockchain, and they also analyzed the security of these systems. In order to solve the insecure problem of three-party authentication, Mamun et al. [17] revised the RFID authentication protocol. Relying on ECC, Kumar et al. [18] proposed an identity authentication system, and used this system for IoT and cloud servers in smart cities. In order to realize the identity authentication of the entity, Bae et al. [19] proposed a smart card-based identity verification protocol. The user can log in to the server of the Internet of Things through the smart card. In response to the shortcomings of the previous scheme, Kumari et al. [20] proposed an ECC-based authentication scheme and proved their security under the Internet of Things and cloud servers.

### *2.2. Hash Chain*

The original purpose of the hash chain is to solve the problem that the encrypted message is easy to steal or be tampered with during transmission. After multiple encryption iterations of the message, the hash chain can effectively resist attacks and interference and can complete verification at the receiving end.

With the development of cryptography in recent years, the hash chain has also been continuously developed. Different forms of hash chains have been proposed by scholars, such as a star hash chain, a tree hash chain and a butterfly hash chain [21]. The main purpose of these different forms of hash chains is to solve communication problems, such as tolerance to packet loss, load, etc. Although the proposal of these hash chains has solved some problems existing in the hash chains, they are still only optimized on the basis of the original hash chains without further expansion. Subsequently, hash chains began to be applied to authentication; however, this scheme has difficulty achieving a balance in the number of iterations of the hash function. Once the hash function is used for too many iterations, the efficiency of authentication will be reduced; and once the number of iterations is too small, the security of authentication will be lost. Huang et al. [22] proposed to use different hash functions to iterate messages. However, this new hash chain construction method needs to ensure that the order of hash functions used during iteration is not stolen. Kim et al. [3] used hash chains in the smart home space, and they proposed a reverse hash chain that can replace PoW in the blockchain and thus be used to verify the identity of users in smart homes.

### 3. Multi-User Collaborative Access Control

This section introduces a multi-user collaborative access control method based on a new hash chain. The method mainly includes four algorithms: (1) Construction algorithm of the new hash chain; (2) PoW algorithm based on the new hash chain; (3) Establishment algorithm of multi-user collaborative access control; (4) Realization Algorithm of Multi-user Collaborative Access Control. The model of the multi-user collaborative access control method is shown in Figure 1 below, and the definitions of abbreviations and symbols are shown in Table 1. This method involves the cloud, blockchain, and edge nodes in total. Through these four algorithms, this method realizes the security authentication of qualified users and the interception of unauthenticated users, and the information transmission between them follows SSL/TLS. In the nodes of the blockchain, each node maintains a hash function library containing the same hash function.

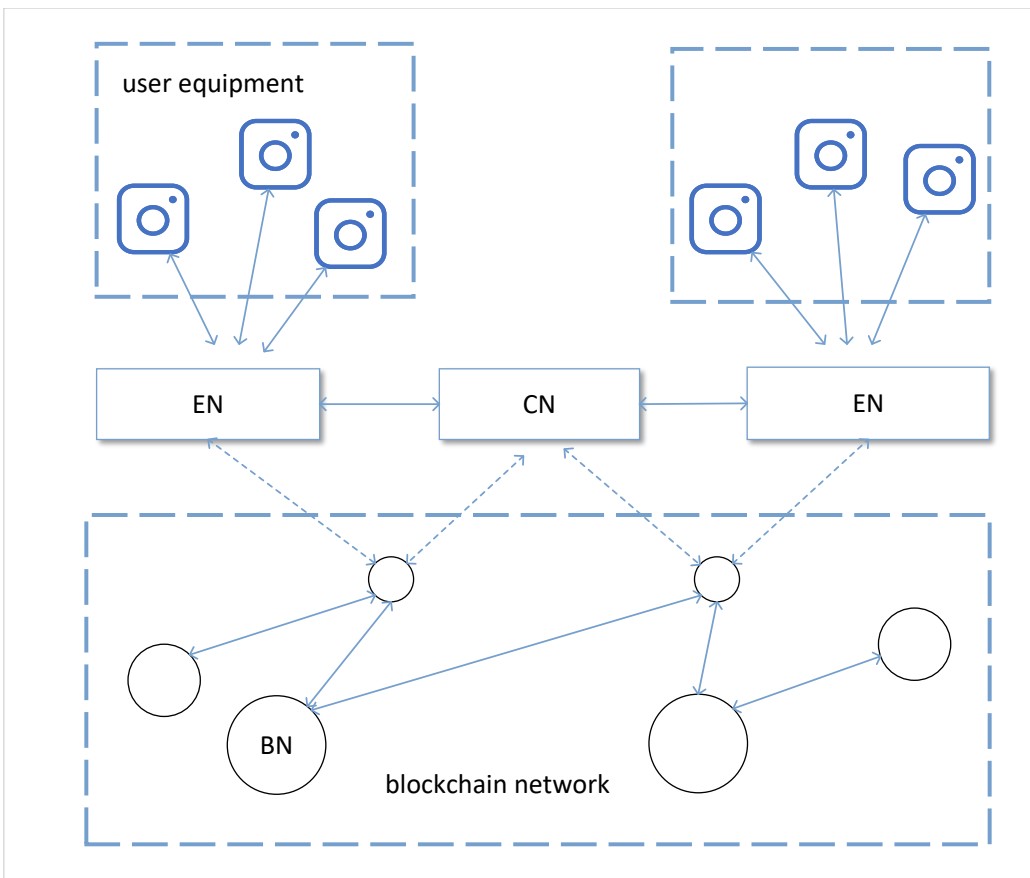

**Figure 1.** The model of the multi-user collaborative access control method.

**Table 1.** Abbreviations and definitions.

| Abbreviations | Definitions |
| --- | --- |
| $EN$ | Edge node |
| $BN$ | Blockchain node |
| $CN$ | Cloud node |
| $U_i$ | The *i*-th identity information of the user |
| $DB_X$ | *X*'s database |
| $SQ_X$ | Sequence of *X* |
| $HCN_i$ | The *i*-th element of the hash chain |
| $PU_X$ | The public key generated by *X* |
| $h_i(X)$ | The hash value generated using the *i*-th hash function |
| $PoW_j$ | Proof-of-work solution for block *j* |
| $Q(X)$ | Request with content *X* |
| $X||Y$ | The parallel operation of *X* and *Y* |
| $E(X)_Y$ | Use *Y* to generate the ciphertext of *X* |
| $SIG(X)_Y$ | Use *Y* to generate the signature of *X* |

### 3.1. Access Control Process

This chapter describes the overall process of a multi-user collaborative access control scheme. The process of the scheme is shown in Figure 2. The specific steps for access control are as follows:

- The edge node $EN$ applies for access from cloud nodes $CN$ and blockchain nodes $BN$;
- After receiving the consent request message from the cloud node, the edge node $EN$ accepts the key sequence $SQ_P$ from the user;
- The edge node $EN$ gets the order $O_{EN}$ in which the hash chain is built from the cloud node $CN$;

- The blockchain node $BN$ accepts user keys $SQ_P$ as well as the build order $O_{EN}$ from all edge nodes, and then builds a new hash chain $HCN_m$ based on sequence $SQ_P$ and order $O_{EN}$;
- The blockchain node $BN$ participates in blockchain consensus based on the hash chain, and builds solution $G_j$ based on the new hash chain $HCN_m$;
- The blockchain node $BN$ determines whether solution $G_j$ satisfies $PoW_j$ and, if not, the access fails; if satisfied, the blockchain node $BN$ broadcasts solution $G_j$ to other blockchain nodes;
- Other blockchain nodes verify solution $G_j$, and if solution $G_{j-1}$ is not equal to $h_k(h_k(s)||G_j)$, the access fails; if they are equal, the consensus is completed and the block is added;
- The edge node $EN$ returns the access result to the user.

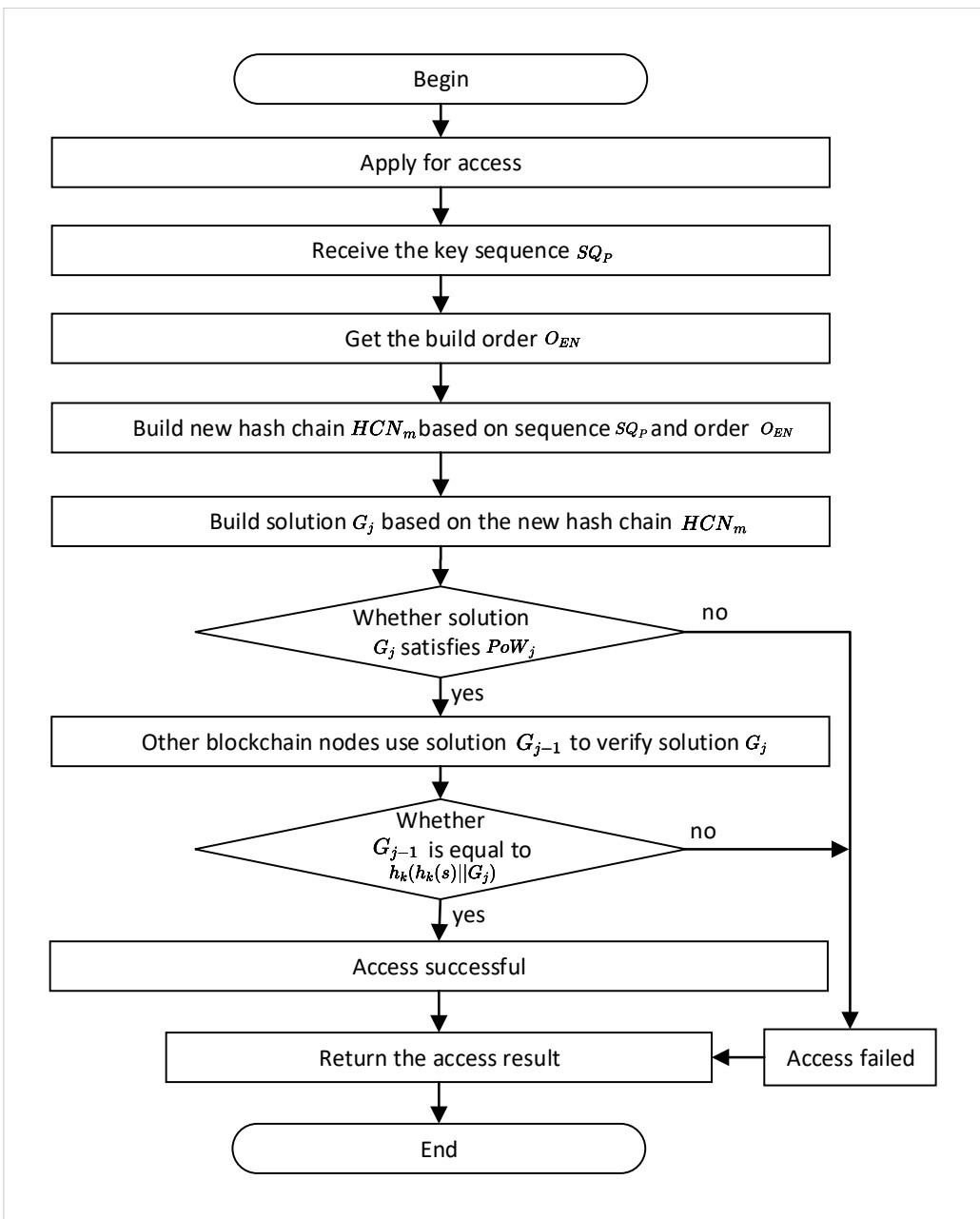

**Figure 2.** Access control flowchart.

### 3.2. Construction Algorithm of New Hash Chain

In this Algorithm, the blockchain node first collects keys according to the order in which the hash chain is constructed, and generates the key sequence $SQ_P = \sum_{i=1}^{m} pswd_i$, where $m$ is the total number of keys, and then, the algorithm processes each key according to the corresponding hash function $h_k$ to obtain the hash sequence $SQ_H = \sum_{i=1}^{m} h_k(pswd_i)$, where $k = i \mod n$, $n$ is the total number of hash functions. Finally, the algorithm uses the hash sequence $SQ_H = \sum_{i=1}^{m} h_k(pswd_i)$ to construct a new type of hash chain. The first hash sequence will be recognized as the seed value of the new hash chain, and then the current hash sequence value and the previous hash chain element is composed to build the next hash chain element. That is, the $i$-th element of the hash chain is $HCN_i = h_k(h_k(pswd_i)||HCN_{i-1})$. The complete set of hash chains constructed is:

$$HCN_i = \begin{cases} HCN_i = h_k(h_k(pswd_i)||HCN_{i-1}) & 1 < i \leqslant m \\ h_1(pswd_1) & i = 1. \end{cases} \tag{1}$$

The specific steps are as follows:

Step 1: The blockchain node $BN$ collects keys according to the order in which the hash chain is constructed, and generates key sequences $SQ_P = \sum_{i=1}^{m} pswd_i$;

Step 2: The blockchain node $BN$ selects a hash function $h_k$ according to the subscript of each value in the sequence, and then uses the corresponding hash function to calculate each key to obtain a hash sequence $SQ_H = \sum_{i=1}^{m} h_k(pswd_i)$;

Step 3: The blockchain node $BN$ uses the first value of the hash sequence $h_1(pswd_1)$ as the seed value of the hash chain to obtain the first element of the hash chain $HCN_1 = h_1(pswd_1)$;

Step 4: The blockchain node $BN$ combines the $i$-th value of the hash sequence with the previous element, and the hash iteration obtains the $i$-th element of the hash chain $HCN_i = h_k(h_k(pswd_i)||HCN_{i-1})$;

Step 5: The blockchain node $BN$ judges whether all the values in the hash sequence have been calculated. If yes, turn to step 6; if not, let $i = i + 1$, and return to step 4;

Step 6: The construction algorithm of the new hash chain ends.

Figure 3 shows the algorithm construction process.

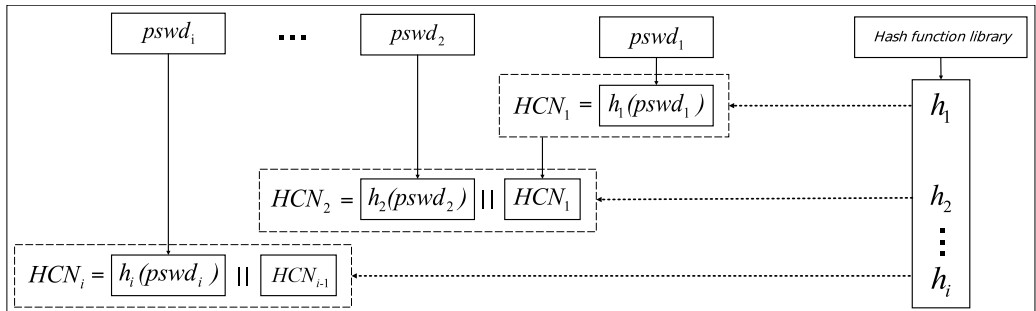

**Figure 3.** The construction diagram of the new hash chain.

### 3.3. Pow Algorithm Based on New Hash Chain

In this algorithm, blocks are issued by completing PoW. Assuming that $PoW_j$ is the solution of the PoW of the $j$-th block, as long as the problem of $PoW_j$ can be solved correctly, the $j$-th block can be generated and the consensus can be completed. Therefore, it is necessary to ensure that only users participating in collaborative access can solve the $PoW_j$. At the same time, it is necessary to ensure that other blockchain nodes can verify the solution under the condition that $PoW_j$ cannot be solved. Let $G(m, l) = \{G_j\} (1 \leqslant j \leqslant l)$ be the solution set of PoW, where $m$ is the number of iterations by the key and $l$ is the effective number of iterations by the salt. The value of $G_j$ is the value after $m$ key iterations and

$l - j$ salt value iterations. That is to say, the $j$-th element of $G(m, l)$ is $G_j = h_k(h_k(s)||G_{j+1})$, where $k = (j + m) \mod n$, and s is the salt value. The corresponding solution of $PoW_j$ is:

$$PoW_j = G_j = \begin{cases} HCN_m & j = l \\ h_k(h_k(s)||G_{j+1}) & l > j \geqslant 1. \end{cases} \quad (2)$$

Due to the single irreversible characteristic of the hash function, for each $G_{j+1}$ to be solved, it is difficult for the blockchain node to reversely solve it through the known $G_j$. However, for each solved $G_j$, the blockchain can verify the solution by easily comparing the values of $G_j$ and $h_k(h_k(s)||G_{j+1})$ for equality. That is to say, the blockchain nodes cannot guess the $PoW_j$ to be solved through the known $PoW_{j+1}$, but can easily verify the solution $PoW_j$ with the solution $PoW_{j-1}$. Therefore, under the condition that only users participating in collaborative access have the key and the valid number of iterations of the salt value, the algorithm calculates the hash chain through the blockchain nodes, and completes the verification on other nodes to complete the issuance of the block.

The specific steps are as follows:

Step 1: The blockchain node $BN$ obtains the serial number $j$ of the block to be generated, and obtains the effective number of iterations $l$ of the salt value.

Step 2: The blockchain node $BN$ uses the hash chain $HCN_m$ constructed by algorithm as the seed value to obtain the first element of the hash chain.

Step 3: After the blockchain node $BN$ combines the hashed salt value $h_k(s)$ with the seed value and completes $l - j$ hash iterations, $G_j = h_k(h_k(s)||G_{j+1})$ is obtained.

Step 4: The blockchain node $BN$ verifies whether the value of $PoW_{j-1}$ is equal to $h_{k-1}(h_{k-1}(s)||G_j)$; if they are equal, turn to step 5 and If not, return to step 2.

Step 5: The blockchain node $BN$ generates $j$-block and adds it to the longest chain, publicly announcing that the block is valid.

Step 6: Other blockchain nodes use the $k - 1$ th hash function and the $PoW_j$ of the block to verify that the value of $PoW_j$ is equal to $h_{k-1}(h_{k-1}(s)||G_j)$. If the verification is successful, agree to go to the chain; If not, refuse to go to the chain.

Step 7: The PoW algorithm based on the new hash chain ends.

Figure 4 shows the algorithm construction process.

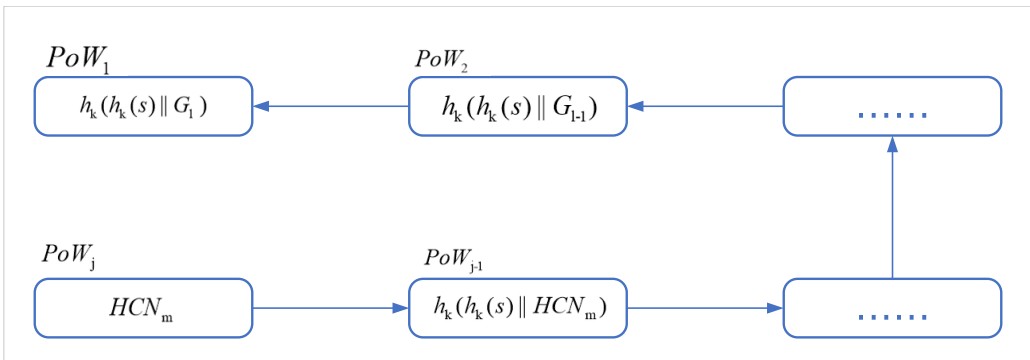

**Figure 4.** The construction diagram of the new hash chain.

### 3.4. Establishment Algorithm of Multi-User Collaborative Access Control

In this algorithm, edge node $EN$ first registers on cloud node $CN$. Cloud node $CN$ completes the authentication of edge node $EN$, then saves the edge nodes participating in the visit in database $DB_{CN}$ according to groups. The cloud node $CN$ waits for all edge nodes to complete registration and returns the registration order of all edge nodes, which is the order of the hash chain. Then, the edge node $EN$ will complete the registration with the blockchain node $BN$. After the blockchain node $BN$ completes the authentication of the edge node $EN$, it will judge whether the valid times $l$ provided by all the edge nodes are the same. After the blockchain node $BN$ waits for the registration of all edge nodes to complete, it constructs a key sequence $SQ_P = \sum_{i=1}^{m} pswd_i$ according to the registration

order, and then constructs a hash chain $HCN_m$ according to the construction algorithm of the new hash chain. Finally, the blockchain node $BN$ solves the algorithm according to the PoW algorithm based on the new hash chain.

The specific steps are as follows:

Step 1: The edge node $EN$ uses the ECC to generate a pair of key pairs $(PU_{EN}, PR_{EN})$, the former one is the public key, and the latter one is the private key. Then the edge node $EN$ signs the message $msg_1$ with the content "apply for registration" to generate signature $SIG(msg_1)_{PR_{EN}}$. Then the edge node $EN$ transmits the registration request signature to the cloud node $CN$.

Step 2: The cloud node $CN$ receives the registration request message from edge node $EN$, then signs the message $msg_2$ with the content "agree to registration" to generate signature $SIG(msg_2)_{PR_{EN}}$. The cloud node $CN$ transmits the response signature and public key $PU_{CN}$ to the edge node $EN$.

Step 3: The edge node $EN$ receives the response message from the cloud node $CN$, then uses the public key of the cloud node to verify the signature. If the edge node $EN$ successfully verifies the signature, turn to step 4; If not, the edge node $EN$ deletes the response information and return to step 1.

Step 4: The edge node $EN$ encrypts the identification number $ID_{EN}$ and public key $PU_{EN}$ to generate encrypted file $E(ID_{EN}||PU_{EN})_{PU_{CN}}$. Then, edge node $EN$ transmits the encrypted information to cloud node $CN$.

Step 5: The cloud node $CN$ receives the registration request message from edge node $EN$, uses its own private key $PR_{CN}$ to decrypt the registration information, and obtains the identification number $ID_{EN}$ and public key $PU_{EN}$ of the edge node. Then, cloud node $CN$ verifies the registration signature. If the cloud node $CN$ successfully verifies the signature, turn to step 6; if not, the cloud node $CN$ deletes the request information and return to step 1.

Step 6: The cloud node $CN$ generates timestamp $T$ and then saves the identification number $ID_{EN}$ and timestamp $T$ to the database $DB_{CN}$.

Step 7: The cloud node $CN$ waits for registration requests from other edge nodes within a limited time $T_\Delta$, and saves the registration information in the database $DB_{CN}$.

Step 8: The cloud node $CN$ judges whether all edge nodes have been registered. If completed, turn to step 9; if not, return to step 1.

Step 9: The cloud node $CN$ generates the serial number $O_{EN}$ of the edge node according to the timestamp, and then encrypts it with its corresponding public key $PU_{EN}$ to generate an encrypted file $E(O_{EN})_{PU_{EN}}$. Then, the cloud node $CN$ sends the encrypted file to the corresponding edge node $EN$.

Step 10: The edge node $EN$ receives the registration request message from cloud node $CN$, decrypts it with its own private key $PR_{EN}$, and obtains the serial number $O_{EN}$. Then, edge node A collects the key $pswd$ and valid times $l$ input by the user.

Step 11: The edge node $EN$ signs the message $msg_1$ with the content "apply for registration", generates the signature $SIG(msg_1)_{PR_{EN}}$, and then uses the public key of the blockchain node $PU_{BN}$ to encrypt the user key $pswd$, valid times $l$, serial number $O_{EN}$, and public key $PU_{EN}$ to generate an encrypted file $E(pswd||l||O_{EN}||PU_{EN})_{PU_{BN}}$. The edge node $EN$ transmits registration request signatures and encrypted information to the blockchain node $BN$.

Step 12: After receiving the registration request message from the edge node $EN$, the blockchain node $BN$ uses its own private key $PR_{BN}$ to decrypt the registration information, and obtains the user key $pswd$, valid times $l$, serial number $O_{EN}$, and the public key of the edge node $PU_{EN}$. The blockchain node $BN$ then verifies the registration signature. If the blockchain node $BN$ successfully verifies the signature, turn to step 13; If not, the blockchain node $BN$ deletes the request information and return to step 9.

Step 13: The blockchain node $BN$ waits for registration requests from other edge nodes within a limited time $T_\Delta$, and judges whether their valid times $l$ are equal. If they are equal, turn to step 14; if not, return to step 12.

Step 14: The blockchain node $BN$ judges whether all edge nodes have been registered. If completed, turn to step 15; if not, return to step 11.

Step 15: The blockchain node $BN$ sorts all user keys according to the serial number to generate a key sequence $SQ_P = \sum_{i=1}^{m} pswd_i$.

Step 16: The blockchain node $BN$ constructs a hash chain according to the construction algorithm of the new hash chain, and obtains a hash chain $HCN_m$ composed of all user keys.

Step 17: The blockchain node $BN$ solves the $PoW_1$ of the block with serial number 1 and effective times according to the PoW algorithm based on the new hash chain.

Step 18: The blockchain node $BN$ generates an initial block and uploads it to the chain, publicly announcing that the block is valid.

Step 19: The establishment algorithm of multi-user collaborative access control ends.

Figure 5 shows the algorithm construction process.

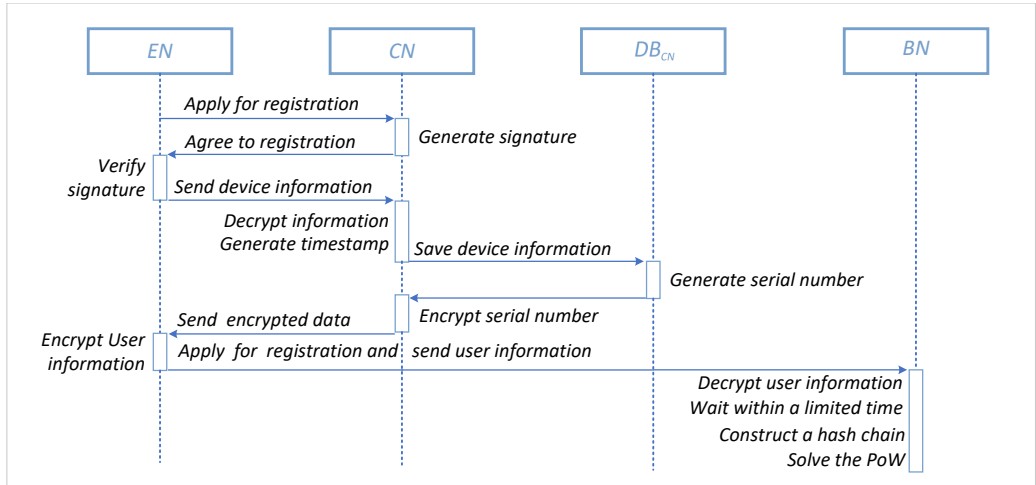

**Figure 5.** The construction diagram of the new hash chain.

### 3.5. Realization Algorithm of Multi-User Collaborative Access Control

In this algorithm, the edge node $EN$ first completes the authentication at the cloud node $CN$ and, after the cloud node $CN$ completes the authentication of the edge node $EN$, it returns the corresponding serial number $O_{EN}$. Then, the edge node $EN$ collects the user's authentication information and applies for access to the blockchain node $BN$. After the blockchain node $BN$ completes the authentication of the edge node $EN$, it obtains the user information participating in access control within a limited time, and judges whether all the effective times $l$ are the same. Then, the blockchain node $BN$ builds the key sequence according to the registration order $O_{EN}$, and then builds the hash chain $HCN_m$ according to the construction algorithm of the new hash chain. Finally, the blockchain node $BN$ solves the algorithm according to the block serial number $j$ and PoW algorithm based on the new hash chain.

The specific steps are as follows:

Step 1: The edge node signs $EN$ and the message $msg_3$ with the content "apply for access". The edge node $EN$ uses the public key of the cloud node $PU_{CN}$ to encrypt the identification number and public key of the edge node $PU_{EN}$ to generate an encrypted file $E(PU_{EN}||ID_{EN})_{PU_{CN}}$. The edge node transmits access request signatures and encrypted information to the cloud node $CN$.

Step 2: The cloud node $CN$ receives the access request message from the edge node $EN$, uses its own private key $PR_{CN}$ to decrypt the authentication information, and obtains the public key $PU_{EN}$ and identification number $ID_{EN}$ of the edge node. The registration signature $SIG(msg_3)_{PU_{EN}}$ is then verified. If the cloud node $CN$ successfully verifies the signature, turn to step 3; if not, the cloud node $CN$ deletes the request information, and return to step 1.

Step 3: The cloud node $CN$ uses the identification number of the edge node $EN$ to query the database. If the query is successful, it will obtain the serial number $O_{EN}$ corresponding to the identification number, and then turn to step 4; if the query fails, print an error and return to step 1.

Step 4: The cloud node $CN$ encrypts the serial number $O_{EN}$ with the public key $PU_{EN}$ of the edge node, and then transmits the encrypted file $E(O_{EN})_{PU_{EN}}$ to the edge node.

Step 5: After receiving the encrypted file from the cloud node $CN$, the edge node $EN$ decrypts it to obtain the serial number. Then, the edge node $EN$ collects the key *pswd* entered by the user and the number of valid times $l$.

Step 6: The edge node $EN$ signs the message with the content "apply for access", generates a signature $SIG(msg_2)_{PR_{EN}}$, and then uses the public key of the blockchain node $PU_{BN}$ to encrypt the user key *pswd*, valid times $l$, serial number $O_{EN}$, and public key $PU_{EN}$ to generate an encrypted file $E(pswd||l||O_{EN}||PU_{EN})_{PU_{BN}}$. The edge node $EN$ transmits registration request signatures and encrypted information to the blockchain node $BN$.

Step 7: After receiving the encrypted information of the edge node $EN$, the blockchain node $BN$ decrypts the information to obtain the user key *pswd*, valid times $l$, serial number $O_{EN}$, and the public key provided by the edge node $PU_{EN}$. The blockchain node then verifies the signature. If the edge node $EN$ successfully verifies the signature, turn to step 8; if not, delete the request information and return to step 6.

Step 8: The blockchain node $BN$ waits for access requests from other edge nodes within a limited time $T_\Delta$, and judge whether their valid times $l$ are equal. If they are equal, turn to step 9; if not, return to step 6.

Step 9: The blockchain node $BN$ judges whether all edge nodes have been authenticated. If completed, turn to step 10; if not, return to step 6.

Step 10: The blockchain node $BN$ sorts all user keys according to the serial number to generate a key sequence $SQ_P = \sum_{i=1}^{m} pswd_i$.

Step 11: The blockchain node $BN$ constructs a hash chain according to the construction algorithm of the new hash chain, and obtains a hash chain $HCN_m$ composed of all user keys.

Step 12: The blockchain node $BN$ obtains the serial number $j$ of the block to be generated, and judges whether the verification is successful according to the number of valid times $l$ and PoW algorithm based on the new hash chain. If successful, turn to step 13; if not, return to step 1.

Step 13: The blockchain node $BN$ signs the message $msg_4$ with the content "agree to access", generates a signature $SIG(msg_4)_{PR_{BN}}$, and then transmits it to the edge node $EN$.

Step 14: After the edge node $EN$ receives the signature from the blockchain node $BN$, it uses the public key of the blockchain node $PU_{BN}$ to verify it. If the signature verification is successful, it will display "Access Successful"; if not, it will display "Access Failed".

Step 15: The realization algorithm of multi-user cooperative access control ends.

Figure 6 shows the algorithm construction process.

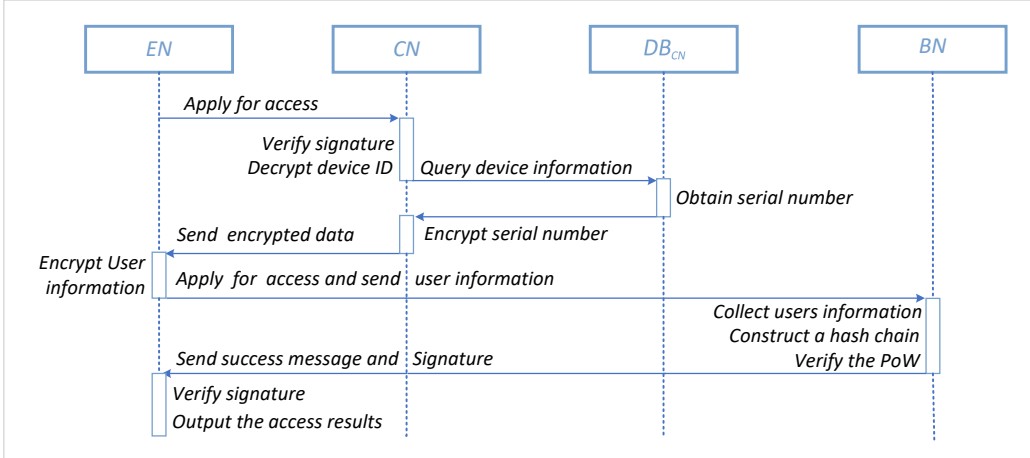

**Figure 6.** The construction diagram of the new hash chain.

## 4. Security Analysis

This article uses a novel hash chain mechanism to securely control data in the field of multi-user collaborative access. Therefore, we mainly focus on the security of authentication with access control. Table 2 below lists the threats related to data authentication that we focus on analyzing.

**Table 2.** Threat List.

| Type of Threat | Detail of Threat |
| --- | --- |
| Type 1 | The fake block is generated |
| Type 2 | The fake block is inserted |
| Type 3 | The fake transaction is generated |
| Type 4 | The block is deleted |
| Type 5 | The block has been tampered with |
| Type 6 | The *BN* is illegally closed |
| Type 7 | The internal data store is exposed |

- Threat type 1: The fake block is generated. For the consensus mechanism in the field of multi-user collaborative authentication, we adopted the same assumption as the general consensus mechanism, that is, the longest blockchain is used for authentication. Therefore, when an attacker creates forged blocks in the local blockchain, the authentication process cannot be affected. Another attack scenario for the attacker is to forge *BC* and generate blocks, but this paper proposes to send the creation information to other blockchain nodes for verification, and other blockchain nodes can recognize that the current forged node is fake.
- Threat type 2: The fake block is inserted. First, the block insertion operation needs to obtain the permission of the blockchain node, and the attacker cannot verify his identity through the certificate, so he cannot obtain the insertion permission. Second, even if a block is inserted illegally, other blockchain nodes can verify the newly inserted block.
- Threat type 3: The fake transaction is generated. Since the generation of the transaction needs to obtain the key provided by the edge node, and the new transaction cannot be reversed through the existing transaction, the attacker cannot successfully forge the transaction. Assume that the key of the edge node is illegally stolen by the attacker, but in the scenario of multi-user collaborative authentication, it is difficult for the attacker to obtain the keys of all nodes participating in the authentication. In addition, in order to further increase the security of access control, block nodes will avoid using the initial value of the key when generating transactions, but will use different

functions to hash and iterate the node key according to certain rules. Therefore, even if an attacker obtains all the original keys, transactions cannot be generated.

- Threat type 4: The block is deleted. Since the scheme is certified by the longest blockchain, even if an attacker deletes a block in the local blockchain, it cannot affect the authentication process. At the same time, the current transaction can be mutually verified with the previous transaction. Therefore, the deletion of any block in the blockchain will be discovered by the blockchain nodes.
- Threat type 5: The transaction has been tampered with. Since the transaction information is generated by the same set of keys according to certain rules, the blockchain nodes can calculate and obtain all possible generated transaction information. In this way, blockchain nodes can determine whether the current blockchain has been tampered with.
- Threat type 6: The blockchain nodes are illegally shut down. If the blockchain nodes in the domain are shut down, the blockchain nodes cannot respond to the access requests of other nodes. However, when the scheme is verified, it can be directly authenticated according to the block, without the authority of the blockchain node. Therefore, when a blockchain node is attacked by Dos, other nodes only need to search for the blocks they need to complete the verification.
- Threat type 7: The cloud nodes expose internal data storage. If the cloud node data are obtained illegally, the attacker can obtain the sequence of the key combination. However, the verification process in this scheme requires three conditions: key, key combination sequence, and key encryption rules. These three parts are, respectively, stored in different nodes, so the verification cannot be completed when only the key combination rules are known.

Next, we analyzed the system's ability to withstand attacks. We collected the most common attacks in IoT and analyzed how the system performs in the face of these attacks. Then we compared our method with Ma's method [2] and Kim's method [3]. Table 3 below lists the security function.

**Table 3.** Security Function.

| Security Function | Our Method | Ma's Method | Kim's Method |
| --- | --- | --- | --- |
| Resistant to DoS attacks | Yes | Yes | Yes |
| Resistant to replay attack | Yes | No | Yes |
| Resistant to sniffing attack | Yes | Yes | Yes |
| Resistant to impersonation attack | Yes | Yes | Yes |
| Resistant to password attack | Yes | No | Yes |
| Multi-user collaborative access control | Yes | No | No |

- DoS Attack: DoS Attack means that an illegal attacker occupies system resources, making it impossible for users to complete access control normally. Suppose an attacker shuts down a blockchain node in a domain through a DoS attack, making the blockchain node unable to respond to access requests from other nodes. However, in this strategy, the access control of users is realized, and authentication can be carried out directly according to the block without the need for the permission of the blockchain node. Therefore, when a blockchain node is attacked by DoS, other nodes only need to search for the block they need to complete the authentication.
- Replay Attack: Replay Attack means that an illegal attacker obtains the authentication information and resends it to the recipient in the same way and format. However, in this method, user access control is implemented by using a new type of hash chain; the passwords are completely different during each authentication. After the method completes an access control, the password used in the access control will be used to verify the next access control. Assume that an illegal attacker illegally intercepts the solution $G_x$ of the target number $x$ currently used for authentication, and the user successfully completes the access control to the system through the solution $G_x$.

At this time, the illegal attacker uses the same method, that is, sends the solution $G_x$ of number $x$ to the blockchain node. After receiving the solution, the blockchain node will judge that the solution cannot be verified and discards the information. Therefore, this method can effectively resist replay attacks.

- Sniffing attack: A sniffing attack is when illegal attackers steal transmission information between devices. However, in this method, the user's identity information sequence used for access control will be constructed into a hash chain in advance, and it will be encrypted using the elliptic curve encryption algorithm. Therefore, in this policy, the transmission information between smart home devices is encrypted with ciphertext, and the effective time of transmission is set. The encryption level of the elliptic curve encryption algorithm is very high. With the existing computer processing power, it is difficult for an attacker to crack the ciphertext within the limited time of transmission. With this method, the transmission of data will be supplemented with digital signatures to verify some processing information, which can further ensure the security of user identity information.
- Impersonation attack: An impersonation attack is when illegal attackers forge fake data to make the device misjudge its identity. However, in this method, the edge end nodes, blockchain nodes, and cloud nodes will use digital signatures for two-way authentication to ensure that neither the sender nor the receiver is a fake node. If the signature cannot be verified, the user data package will be deleted. In addition, in the access control implementation algorithm of smart home sensing information, the blockchain node will use the saved solution to verify the target solution, and the attacker cannot complete the verification through the forged solution.
- Password attack: Password attack means that illegal attackers complete access control by guessing user passwords. However, in this method, access control is through the verification of user identity information, which consists of various types, including biometric information, digital tokens, and so on. It is difficult for attackers to guess user identity information through password attacks. In addition, the process of implementing access control requires three conditions: key, key combination sequence, and key encryption rules, and these three parts are stored in different nodes. Even if an attacker guesses and obtains user identity information, access control cannot be completed without knowing the combination rules of identity information.

## 5. Cost Analysis

This section analyzes the time overhead required by the access control strategy, mainly analyzing the time consumed by the hash function and the ECC algorithm, and ignoring the time overhead of information transmission so as to avoid the uncertainty of information transmission time caused by network bandwidth influence. Let the iteration time cost of forward hashing be $T_H$, the time cost of encryption in ECC algorithm be $T_{E1}$, the time cost of decryption in the ECC algorithm be $T_{D1}$, the time cost of the digital signature in the ECC algorithm be $T_{E2}$ and the time cost of verifying the signature in the ECC algorithm. The overhead is $T_{D2}$, the number of blockchain nodes is $n$, the amount of user identity information is $m$, the iteration length is $x$, the number of blocks is $y$, and the time overhead for sorting user identity information once is $T_S$. Table 4 shows the time overhead of each algorithm.

**Table 4.** Comparison of time consumption.

| Algorithm | Time Overhead |
|---|---|
| Construction Algorithm of New Hash Chain | $mlogmT_S + (2m-1)T_H$ |
| PoW Algorithm Based on New Hash Chain | $2(x-y+n)T_H$ |
| Establishment Algorithm of Multi-user Collaborative Access Control | $3T_{E1} + 3T_{D1} + 4T_{E2} + 4T_{D2}$ |
| Realization Algorithm of Multi-user Collaborative Access Control | $4T_{E1} + 4T_{D1} + 4T_{E2} + 4T_{D2}$ |

The traditional blockchain system uses a large number of nodes to continuously calculate and solve PoW. PoW is generally a hash value composed of hexadecimal numbers, and the node may perform calculations that require hundreds of hash functions to complete the PoW solution. In the worst case, such a calculation would take a conventional computer a trillion hours. However, our proposed strategy does not need to solve the hash value backwards, but only requires multiple forward hash iterations, so the overhead required is multiple forward hashes. We compared our scheme with related work, and the results are shown in the Table 5 below.

**Table 5.** Scheme comparison with related work.

| Scheme | Decision Consensus | Time Complexity | Security |
| --- | --- | --- | --- |
| Kim's Method | POW | $O(n)$ | Medium |
| Ma's Method | PBFT | $O(n^2)$ | Strong |
| Our Method | POW | $O(n)$ | Strong |

From the above table, we can conclude that our scheme has higher security than Kim's scheme, and our scheme has less time complexity than Ma's scheme. At the same time, our scheme can be applied to the field of multi-user cooperative access control, and has higher practicability. As the amount of user identity information increases, the security of our scheme is relatively improved. As the number of blocks continues to increase, the number of required hash iterations also decreases, further reducing the time consumption required by our scheme.

## 6. Conclusions

In order to achieve multi-user collaborative access control and solve the limitations of the blockchain in this field, a multi-user collaborative access control scheme based on a new hash chain was proposed. The method mainly includes four algorithms, namely the construction algorithm of the new hash chain, the proof-of-work algorithm based on the new hash chain, the establishment algorithm of multi-user collaborative access control, and the realization algorithm of multi-user collaborative access control. To analyze the reliability of access control methods, we conducted a series of security analyses. Compared with traditional access control schemes, our scheme has higher security and can resist many kinds of attacks. For example, it can resist the generation of counterfeit blocks, data leakage and other threats. In addition, we compared this scheme to the traditional blockchain scheme and conducted a performance analysis of this scheme; the time overhead required by this scheme is much smaller than that of the traditional blockchain. Therefore, this scheme is suitable for environments with limited IoT resources and low computing power.

**Author Contributions:** Conceptualization, Z.W. and D.Z.; methodology, Z.W.; software, D.Z.; validation, Z.W., Y.L. and D.Z.; formal analysis, Z.W.; investigation, D.Z. and G.L.; resources, Z.W. and G.L.; data curation, Z.W. and G.L.; writing—original draft preparation, Z.W.; writing—review and editing, Z.W.; visualization, Z.W. and G.L.; supervision, Z.W.; project administration, Z.W.; funding acquisition, Y.L. All authors have read and agreed to the published version of the manuscript.

**Funding:** This research received no external funding.

**Institutional Review Board Statement:** Not applicable.

**Informed Consent Statement:** Not applicable.

**Data Availability Statement:** Not applicable.

**Conflicts of Interest:** The authors declare no conflict of interest.

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
