# Peer review of "A Multi-User Collaborative Access Control Scheme Based on New Hash Chain"

_electronics, doi:10.3390/electronics12081792_

Round 1

Reviewer 1 Report

In this paper, the authors proposed a multi-user collaborative access control scheme  based on a new hash chain, that uses the identity information of multiple users as the seed value to  construct the hash chain, and uses the hash chain as the PoW of the blockchain.  However, I found the following deficiency in the article, which the authors must rectify:

1. The formal security analysis is missing, which is mandatory for testing the security toughness of any scheme against cyber-attacks.

2. I have not found performance analysis in terms of computation and communication costs and their comparison to existing methods.

3. Also, a comparison of security characteristics with the relevant existing schemes is missing. 

4. In the introduction section,  I was unable to find any citations. The authors should cite relevant articles here from the same field.

Reviewer 2 Report

The authors are offering a multi-user collaborative access control scheme. Indeed, this work is very intersting, but some remarks and correction are required to provide more insights and details on their proposed solution.

The state of the art requires more investigations. The authors are only stating others work but they don’t define their limitations or disadvantages.

There is no deep justification or identification of the limitation of the state of the art. What does the proposed work bring as alternative or innovative solution compared to the SoA.

How the author presented their work is confusing. Only stating how it works in steps is not well elaborated. Better to verify a proper way to describe the working principle.

The cost analysis are not clear, how good or efficient is the proposed system is not clear. Which requirements were indicated, which application assumptions are considered are all missing. How the cost is simulated and evaluated.

The evaluation of the proposed scheme is not well defined, more evaluation are required, what about the security, the response time, the energy. Many others could be investigated. Especially in the abstract, the authors said that their proposed scheme requires small amount of hash value and is good for low computing power system.

The charecterisicts of the proposed scheme are not indicates: How much is the maximum number of user is supported? What about the synchronization to access the chain? 

The comparison to the state of the art is missing.

Round 2

Reviewer 2 Report

The authors have tried to answer the previous questions, however, a thorough and detailed explanation of the proposed method is still lacking. The document needs even more clarification and investigation to bring it to light.

The proposed scheme is based on a theoretical analysis, which makes it a little obscure. It is therefore empirical that the authors propose a clear and well-structured methodology.

In particular, this article necessarily requires an additional comprehensive research flowchart.

Author Response

Please see the attachment. We hope that this revision will be approved

Round 3

Reviewer 2 Report

Thank you for answering the previous remarks. However, still some adjustments.

Figure 2 and text from 141 to 161 are the same.  The text only reports the image in another manner. Please fix it.

Please discuss carefully, the requirments of your scheme in terms of complexity, response time, security and implemnetation efforts.
